# Effects of Anthropogenic Habitat Fragmentation on the Genetic Connectivity of the Threatened and Endemic *Campylorhynchus yucatanicus* (Aves, Troglodytydae) in the Yucatan Peninsula, Mexico

Anay Serrano-Rodríguez [1], Griselda Escalona-Segura [1,*], Antonio González Rodríguez [2], Salima Machkour-M'Rabet [3], Lorena Ruiz-Montoya [4], Eduardo E. Iñigo Elias [5,†] and Alexis Herminio Plasencia-Vázquez [6]

1   El Colegio de la Frontera Sur, Departamento de Conservación de la Biodiversidad, Unidad Campeche, Avenida Rancho Polígono 2-A, Ciudad Industrial, Lerma, Campeche 24500, Mexico
2   Instituto de Investigaciones en Ecosistemas y Sustentabilidad, UNAM, Antigua Carretera a Pátzcuaro No. 8701. Col. Ex Hacienda de San José de la Huerta, Morelia 58190, Mexico
3   El Colegio de la Frontera Sur, Departamento de Conservación de la Biodiversidad, Unidad Chetumal, Avenida Centenario km 5.5, Chetumal 77014, Mexico
4   El Colegio de la Frontera Sur, Departamento de Conservación de la Biodiversidad, Unidad San Cristóbal, Carretera Panamericana y Periférico Sur S/N, Barrio María Auxiliadora, San Cristóbal de Las Casas C.P., Chiapas 29290, Mexico
5   Cornell Lab of Ornithology, Cornell University, Ithaca, NY 14853, USA
6   Centro de Investigaciones Históricas y Sociales, Universidad Autónoma de Campeche, Avenida Agustín Melgar, Campeche 24039, Mexico
*   Correspondence: gescalon@ecosur.mx
†   Retired.

**Abstract:** Identifying connectivity patterns among remnant bird populations and their relationships with land use practices and adjacent habitat fragments is key to implementing appropriate long-term management strategies for species conservation. The coastal scrub and dune vegetation complex of the northern Yucatan Peninsula is rich in endemisms and has been affected by human development, which threatens the survival of the Yucatan Wren (*Campylorhynchus yucatanicus*) population, an endemic bird species. To identify possible anthropogenic barriers to the connectivity of *C. yucatanicus* along 14 localities in the Yucatan (Mexico) coastal north, we explored the relationship between the species population's genetic variability at each sampled site and landscape structure using regression models, in addition to the relationship between genetic distance and landscape resistance. Seven nuclear microsatellite loci were used as genetic markers. Four genetic populations were highlighted by the clustering method implemented in the Geneland program. Human settlement and availability of adequate habitat were significantly related to genetic distance (Fst), suggesting limited connectivity among sites due to ongoing land use changes. We suggest changing the IUCN threat category of *C. yucatanicus* to endangered as we found a significant loss of genetic variability in addition to restricted distribution, small population, habitat degradation, and loss of connectivity.

**Keywords:** conservation; endangered; landscape; microsatellites; Yucatan Wren

## 1. Introduction

Fragmentation and loss of native habitats involve a significant reduction in local and regional biodiversity [1,2] and may alter original species distribution and behavioral responses [3,4]. These processes affect population density [5] and lead to the local extinction of populations [1,6–8]. Factors that cause habitat fragmentation can be both natural and anthropogenic, and both need attention under a climate change scenario to mitigate them and conserve biodiversity. To contain and reverse processes that negatively affect vulnerable

species, it is necessary to understand connectivity patterns and isolation levels among remaining populations of such species to propose appropriate management practices [9–11]. Extended periods of isolation among previously interconnected populations may have serious consequences for species survival because they loss genetic diversity and gene flow [12] and increase inbreeding depression [13,14]. Management for growing the size of local populations could mitigate these problems [15] but preserving or restoring habitat and connectivity between populations could be more beneficial for long-term species survival.

Deforestation and ecosystem fragmentation is extensive in Mexico [16]. Since pre-Hispanic times, the Yucatan Peninsula ecosystems have been subjected to large land use changes by clearing or inducing fires to the native vegetation for crops and cattle ranching, irrigation, water retention, and tourism development [17–23]. The vegetation types most affected are deciduous forest and coastal dune vegetation [24].

The Yucatan Wren (*C. yucatanicus*) (Passeriformes, Troglodytidae) is an endemic bird species native to the extreme northern coast of the Yucatan and a small portion of the Campeche states of Mexico that inhabits almost exclusively the coastal thorn scrub forest and coastal dune vegetations [17,25–31]. The survival of this species is of great concern to Mexico and the world because it is restricted to this fragile and rare ecosystem in the region [32,33]. It is currently unknown how *C. yucatanicus* populations are responding to changes caused in their habitat after years of natural and anthropogenic impacts and the current climate change scenario. Although in the federal legislation of Mexico, "NOM-059-2010", *C. yucatanicus* is listed as "endangered" [34], IUCN reports it only as "near threatened", a lower risk category [35].

Threatened species with low dispersion rates living in restricted and fragmented habitats, such as *C. yucatanicus*, may experience reductions in population sizes and connectivity [36–38]. Identifying population status and connectivity patterns and their association with underlying land-use practices is fundamental to implementing appropriate management strategies for biodiversity conservation [8–11].

In this study, we aassessed the levels and distribution of genetic diversity within and among *C. yucatanicus* populations, which were defined a priori by the distribution of their habitat and the presence of a relatively large number of individuals in our study sites. Our specific objectives were to (i) describe the structure and genetic diversity of *C. yucatanicus* populations of the northern Yucatan Peninsula and identify possible anthropogenic barriers to gene flow, (ii) explore the relationship between the genetic variability of *C. yucatanicus* populations and landscape structure, (iii) describe the relationship between genetic distance and landscape resistance, and finally (iv) propose strategies to increase the genetic connectivity of *C. yucatanicus* in the remaining habitat fragments for this species.

## 2. Methods

### 2.1. Study Area and Sample Collection

Fieldwork was performed through an intensive survey of individuals throughout the range of the species from March 2015 to February 2016. We visited sites with historical records of presence with coastal scrub or dune vegetation complexes with mangrove edges. Sites where individuals of the species were no longer found were discarded; thus, the fieldwork included 14 sites (Figure 1).

To locate individuals to be potentially sampled at each site, during three days from 07:00 to 11:00 and from 16:00 to 19:00 h, we walked transects of variable length (from 1 to 5 km), depending on the extent of vegetation favorable to the species. In total, we surveyed 1077.58 km where the species could be found. At each site, we first walked the transects searching for *C. yucatanicus*; if we obtained no sighting records, we used playbacks every 100 m for 6 min. For this, we downloaded xeno-canto vocalizations of *C. yucatanicus* (https://xeno-canto.org/; accessed on 2 February 2015).

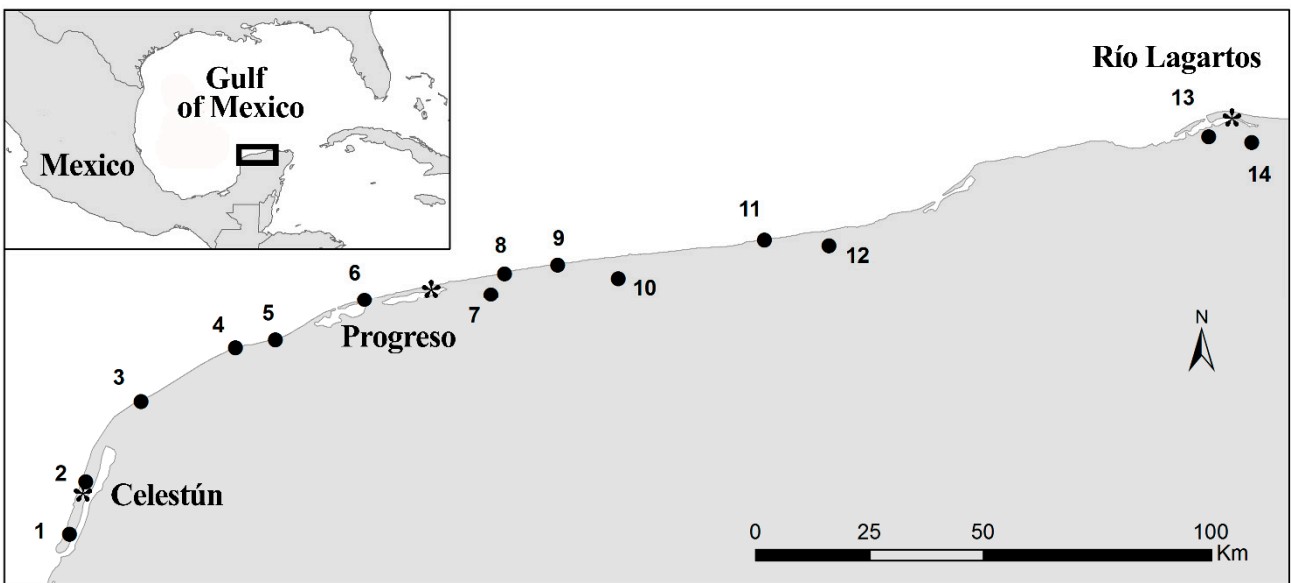

**Figure 1.** Sampled sites in the northwest and central tip of the Yucatan Peninsula, Mexico to study the genetic diversity of *Campylorhynchus yucatanicus* populations in the field between 2015 and 2016: (1) southwest Celestún, (2) northeast Celestún, (3) El Palmar, (4) west Sisal, (5) east Sisal, (6) Chuburná, (7) Capilla, (8) west Chixchulub, (9) San Benito, (10) Xcambó, (11) Santa Clara, (12) Dzilam, (13) west Ría Lagartos, and (14) east Ría Lagartos. We locate the main coastal cities (*).

When we recorded individuals of the species, we set up two adjacent nets of 12 by 3 m with a mesh of 2.5 cm. After capturing one or more individuals and taking the necessary data, we released the individuals and continued the search for a new individual along the transect. Since this species is territorial, setting the nets in a single location would probably have resulted in the repeated capture of the same individuals.

Each captured individual was marked with a unique combination of colored Darvick leg bands to avoid recaptures. We extracted blood samples (2–3 µL) from the brachial vein [39]. We ensured that individuals did not present bleeding when they were released. We collected blood in tubes with K3EDTA at 15%, and samples were stored at −20 °C until processing in the laboratory. Access to protected areas and collections was granted by the SGPA/DGVS/06821/14 and SGPA/DGVS/007765/15 permits.

*2.2. Laboratory Process*

We extracted DNA using a modification of the cell lysis method and phenol-chloroform isoamyl alcohol [40]. Successful DNA extractions were detected by electrophoresis in 1.5% agarose gels with SYBR Gold Staining at a constant voltage of 100 V for 30 min. Samples were genotyped using seven microsatellites described by Barr et al. [41] for *C. brunneicapillus* and standardized for this study for *C. yucatanicus* (Table A1). For amplification, we used a mixture containing 1 µL of DNA, 3 µL of master mix Taq DNA Polymerase (Invitrogen™, Waltham, MA, USA), 2.7 µL of ultrapure water, and 0.3 µL of primer, reaching a final volume of 6 µL. We preheated PCR reactions at 94 °C for 3 min and then performed 39 cycles with the following steps: denaturation at 94 °C for 1 min, alignment at a specific temperature for each primer (Table A1) for 1 min, and extension to 72 °C for 1 min. We maintained extension at 72 °C for 10 min and allowed cooling up 10 °C. We visualized PCR products by electrophoresis in 2% agarose gels with SYBR Gold Staining at a constant voltage of 100 V for 30 min. Sizing was conducted by capillary electrophoresis in an Applied Biosystems automatic sequencer, using LIZ-600 as the internal size standard. We analyzed electropherograms using Peak Scanner 1.0 (Applied Biosystems, Waltham, MA, USA).

### 2.3. Genetic Diversity Analysis

We estimated genetic diversity through allele richness (Na) and Shannon diversity index (I) and expected (He) heterozygosity using Genelex 6.5 [42]. We conducted an analysis to determine whether populations had experienced a recent bottleneck in the Bottleneck program [43] using the Wilcoxon test. To do this, we evaluated our data, with a 70% ratio explained by a Step Mutation Model (SMM) and a 30% ratio explained by the Infinite Alleles Model (AMI) in a two-phase model (TPM), as recommended for microsatellites analysis [43].

### 2.4. Genetic Structure Analysis

We determined whether genotypic frequencies at each locus were under Hardy-Weinberg equilibrium (HWE) and evaluated linkage disequilibrium (LD) among pairs of loci in Genepop 4.6 [44]. To characterize the genetic structure of the species, we used several methods. Bayesian clustering analysis was applied to identify genetic groups, without prior assignment of individuals to a given population, in Structure 2.3 [45]. To determine the most probable value of *K*, which could be interpreted as the optimal number of genetic groups or true clusters, we ran *K* from 1 to 14, with 10 simulations for each *K* using 10,000 iterations before beginning analysis and 50,000 iterations in the Markov Chain Monte Carlo (MCMC). We used the method proposed by Evanno et al. [46] to define the value of *K*, considering the distribution of Δ*K* in Structure Harvester [47].

Bayesian clustering analysis was also performed in the Geneland 4.0 library [48–51] for R 3.4.1 [52], in which we assumed a spatial model with alleles not correlated. For this analysis, we performed $1 \times 10^6$ MCMC iterations. We considered several populations or groups (*K*) from 1 to 14 and 1 iteration was saved every 100. Uncertainty of 100 m for the geographic coordinates was assumed and we evaluated the MCMC convergence using 10 repetitions in each analysis.

Analysis of molecular variance (AMOVA) allowed for the evaluation of the percentage of variation between genetic groups identified by Structure Harvester and among sites in the same group and individuals. This analysis was carried out in Genelex 6.5 [53].

To explore the possible historical demographic changes in the species, Approximate Bayesian Computation (ABC) was used with the software DIYABC Random Forest v1.0.14 [54]. The scenarios tested were as follows: (i) constant effective population size through time, (ii) recent demographic expansion, (iii) recent bottleneck, (iv) historical demographic expansion followed by a bottleneck, and (v) historical bottleneck followed by a recent demographic expansion (Figure 2; Table A2). For training, 100,000 simulated datasets were run per scenario. Five hundred trees were used for model choice and parameter estimation. The selection of the best scenario was based on linear discriminant and partial least squares analysis.

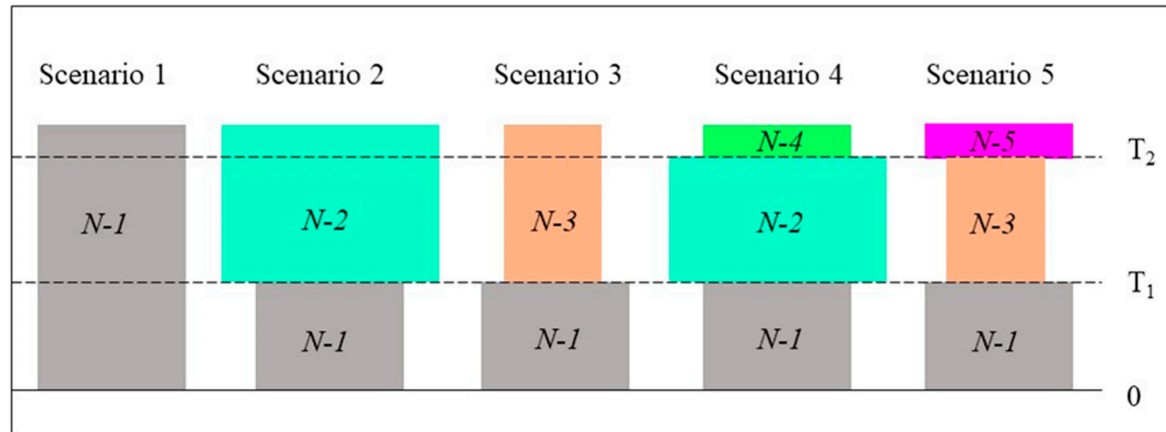

**Figure 2.** Five historical demographic scenarios for *Campylorhynchus yucatanicus* using DIYABC RF. Parameter conditions were as follows: t2 > t1; N2 > N1; N3 < N1; N4 < N2; N5 > N3.

### 2.5. Landscape Composition and Genetic Diversity: Node Level

We defined 14 plots that resulted from a buffer area of 2 km around capture points at each site. To characterize the landscape composition and structure, we used three images of the Sentinel 2 satellite (20 m spatial resolution, April 2016). We manually digitized patches of known vegetation previously confirmed during bird mist netting, which were classified into one of four classes for different types of vegetation and land use: (1) adequate primary habitat preserved, (2) disturbed habitat, (3) unsuitable habitat, and (4) secondary vegetation habitat with some human intervention. The first category consisted of conserved fragments with coastal thorn scrub forest and dune vegetation complexes [32]. Disturbed habitats included human settlements, roads, and areas with bare soil because of deforestation. The third category included habitats where *C. yucatanicus* had not been registered according to the literature [17,26,30,31] and our field observations. Finally, in the fourth category, we considered secondary vegetation that maintained elements of the original vegetation of coastal thorn scrub forests such as agaves and cacti [32] and that was currently being subjected to different human activities, for example, uses with livestock.

The structure and composition of landscapes and their degree of fragmentation were recorded for each plot [7] in Patch Analyst [53]. We selected the following variables: Shannon's patch equitability index (SEI), proportion of suitable habitat (CA1), patch diversity index (SDI), proportion of disturbed habitat (CA2), risk index for proximity to human settlements (PA), edge density of the appropriate habitat patches (ED1), average form index of suitable habitat patches (MSI14), distance to human settlements (SA), distance to road (SC), number of patches of suitable habitat (NumP14), and average size of patches of suitable habitat (MedPS14). We calculated the risk index by proximity of human settlements (PA) with the following formula: PA = PobTotal/S × 100, where PobTotal is the total population of the nearest human settlement and S is the distance to the settlement.

We proposed a priori 12 models to describe the relationship between the genetic diversity of *C. yucatanicus* populations and the configuration of the landscape. These models were evaluated through Akaike Information Criterion (AIC) [55]. The hypotheses proposed were based on the previous knowledge obtained in the literature and our field observations. Models were constructed using linear regressions in R [52]. The expected heterozygosity (He) of *C. yucatanicus* in the sampling sites constituted our response variable, while the predictive or explanatory variables were those obtained in Patch Analyst.

We selected the model with the lowest AIC [55] because it corresponded to greater support of the data. The difference between the AIC value of the best model and that of the remaining models (Δ AIC) made it possible to evaluate their relative hierarchical organization. Those models with ΔAIC < 2 were considered better and equally competitive. Models with 2 < ΔAIC < 4 were considered partially informative. We calculated Akaike's weight ($\omega i$) [55,56] to evaluate the relative likelihood of models being plausible. Regression coefficients ($R^2$) were calculated to verify the fit of data and for models with more than one parameter, an adjusted $R^2$ was taken into account. We calculated *F* statistics and *p* values to determine the statistical significance of regressions.

### 2.6. Genetic Distances and Landscape Resistance

We analyzed the relationship between genetic distance and landscape resistance between sites through linear regression. We also explored the relationship between genetic distance and the spatial Euclidean distance between sites. Calculation of the $F_{ST}$ statistic was performed in Genalex 6.5 [42] and linearized according to the formula $F_{ST}$ mod = $F_{ST}/(1 - F_{ST})$ [57]. We selected $F_{ST}$ mod for the following connectivity analyses because it is the most suitable for these analyses and showed significant correlations, in contrast with the other calculated variables that described the genetic distance between sites, such as the coefficient of genetic differentiation between populations ($G_{ST}$).

The resistance offered by landscape to the movement of individuals was evaluated using circuit theory [58] in Circuitscape [59]. The term "resistance" was used as an antonym for "landscape connectivity", which was defined as the degree to which landscape facili-

tated the movement of individuals [60] that could move randomly between two sites or nodes. In this context, nodes could be habitat fragments, populations, or points in the landscape among which connectivity could be evaluated [59]. We used the centroid of capture points at each collection site to establish focal nodes in this study. The map of the Yucatan Peninsula that contained the *C. yucatanicus* potential distribution allowed for the acquisition of more accurate resistance models [61]. The resistance layer used to estimate the cost or resistance between nodes was used in previous studies for the same species [62].

## 3. Results

We captured 186 individuals of *C. yucatanicus*, but only 130 were included in this study. We excluded individuals with incomplete genetic information during the lab process and juveniles captured with an adult as they moved in familiar groups to minimize genetic similarity due to close kinship.

### 3.1. Genetic Diversity

Genetic variation levels in *C. yucatanicus* differed among sampling sites (Table 1). Allelic richness (Na) turned out to be smaller in west Ría Lagartos, southwest Celestún, Chixchulub, and West Sisal, whereas the Chuburná site presented the highest Na value. Regarding heterozygosity (He), west Ría Lagartos was less diverse. West Sisal and Celestún also had relatively low He values, while San Benito, Xcambó, El Palmar, and Chuburná were the most diverse in terms of He. Our results were significant for the presence of genetic signals of bottlenecks in two sampled sites: San Benito ($p = 0.04$) and Dzilam ($p = 0.04$).

**Table 1.** *C. yucatanicus* genetic variability in 14 sampled sites in the northern Yucatan Peninsula, Mexico. Sample size (N), allele richness (Na), inbreeding index (F), Shannon diversity index (I), expected heterozygosity (He), and standard error (SE).

| ID | Sites | N | Na ± SE | F ± SE | I ± SE | He ± SE |
|----|-------|---|---------|--------|--------|---------|
| 1 | Southwest Celestún | 14 | 2.71 ± 0.28 | 0.08 ± 0.13 | 0.65 ± 0.13 | 0.37 ± 0.07 |
| 2 | Northeast Celestún | 7 | 3.57 ± 0.29 | 0.16 ± 0.07 | 0.72 ± 0.10 | 0.39 ± 0.06 |
| 3 | El Palmar | 9 | 3.14 ± 0.26 | 0.27 ± 0.16 | 0.89 ± 0.05 | 0.52 ± 0.02 |
| 4 | West Sisal | 10 | 2.71 ± 0.42 | 0.21 ± 0.15 | 0.59 ± 0.15 | 0.33 ± 0.08 |
| 5 | East Sisal | 11 | 3.71 ± 0.28 | −0.10 ± 0.05 | 0.89 ± 0.03 | 0.49 ± 0.01 |
| 6 | Chuburná | 14 | 4.14 ± 0.45 | −0.06 ± 0.07 | 0.95 ± 0.12 | 0.50 ± 0.05 |
| 7 | Capilla | 8 | 3.57 ± 0.57 | 0.02 ± 0.09 | 0.81 ± 0.16 | 0.43 ± 0.08 |
| 8 | Chixchulub | 5 | 2.57 ± 0.29 | −0.27 ± 0.11 | 0.73 ± 0.14 | 0.43 ± 0.08 |
| 9 | San Benito | 11 | 3.85 ± 0.49 | −0.06 ± 0.05 | 1.04 ± 0.16 | 0.57 ± 0.06 |
| 10 | Xcambó | 10 | 3.57 ± 0.48 | 0.02 ± 0.16 | 0.95 ± 0.12 | 0.53 ± 0.05 |
| 11 | Santa Clara | 9 | 3.00 ± 0.31 | −0.23 ± 0.04 | 0.75 ± 0.11 | 0.43 ± 0.06 |
| 12 | Dzilam | 9 | 3.71 ± 0.28 | −0.07 ± 0.10 | 0.88 ± 0.10 | 0.48 ± 0.06 |
| 13 | West Ría Lagartos | 3 | 2.42 ± 0.20 | −0.23 ± 0.04 | 0.61 ± 0.09 | 0.36 ± 0.05 |
| 14 | East Ría Lagartos | 10 | 2.85 ± 0.26 | −0.17 ± 0.09 | 0.73 ± 0.11 | 0.41 ± 0.06 |

### 3.2. Genetic Structure

The output of the Bayesian clustering approach implemented in the Structure software indicated that the most probable number of clusters was four (*K* = 4, Table A3, Figure A1). However, the assignment of individuals to four groups in analyzed sites to four groups did not show an evident structure of sampling sites (Figure 3). The groups from the Ría Lagartos sites (east and west, numbers 13 and 14, respectively) appeared to be the most different, considering allocation percentages, and hereafter are collectively called the Ría Lagartos group.

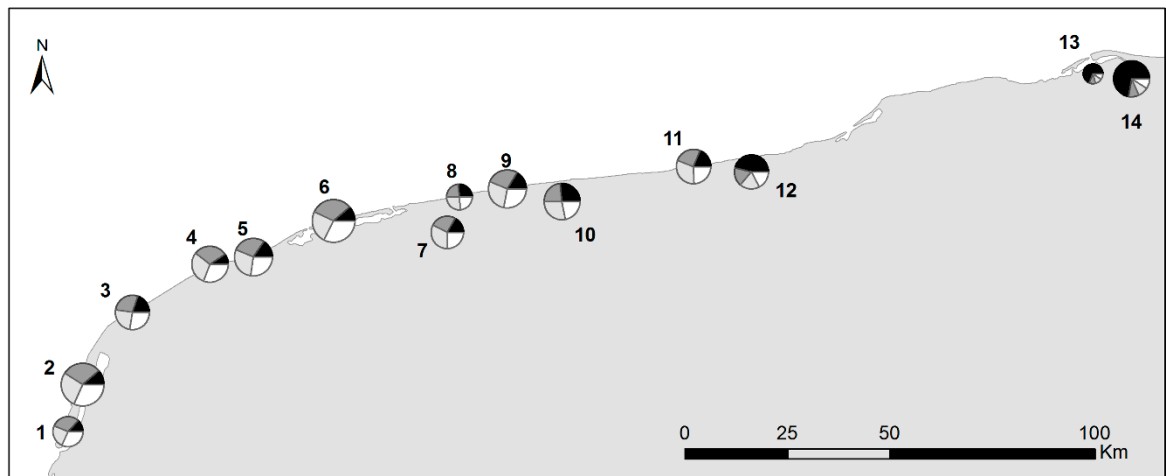

**Figure 3.** Assignment of sampling sites to four genetic groups identified in Structure Harvester (K = 4) of *C. yucatanicus* in the Yucatan Peninsula, Mexico. Sampling sites are as follows: (1) southwest Celestún, (2) northeast Celestún, (3) El Palmar, (4) west Sisal, (5) east Sisal, (6) Chuburná, (7) Capilla, (8) west Chixchulub, (9) San Benito, (10) Xcambó, (11) Santa Clara, (12) Dzilam, (13) west Ría Lagartos, and (14) east Ría Lagartos.

The Bayesian grouping method implemented in Geneland allowed for the differentiation of four genetic groups or populations, again by detecting genetic discontinuities between sampling sites. We obtained Voronoi diagrams that contained the sampling sites grouped in four genetic populations (Figure 4), with an average genetic differentiation (average $F_{ST}$) between them of 0.15 (0.03 < average $F_{ST}$ > 0.28). The group represented by the dark green area with triangles was the most differentiated population (Figure 4) and included individuals captured in Capilla, Chixchulub, and San Benito, with an $F_{ST}$ average of 0.22. The next most differentiated population was from the Ría Lagartos sites (light green area with pentagons in Figure 4), with an $F_{ST}$ of 0.15. In contrast, the lowest $F_{ST}$ average values (orange area with squares in Figure 4) were found in populations in Xcambó, Santa Clara, and Dzilam (average $F_{ST}$ = 0.10). The last identifiable group comprised the largest number of sites located in the west of Progreso city (grey area with circles in Figure 4; average $F_{ST}$ = 0.11). Progreso city is located between the sampling sites Chuburná and Capilla.

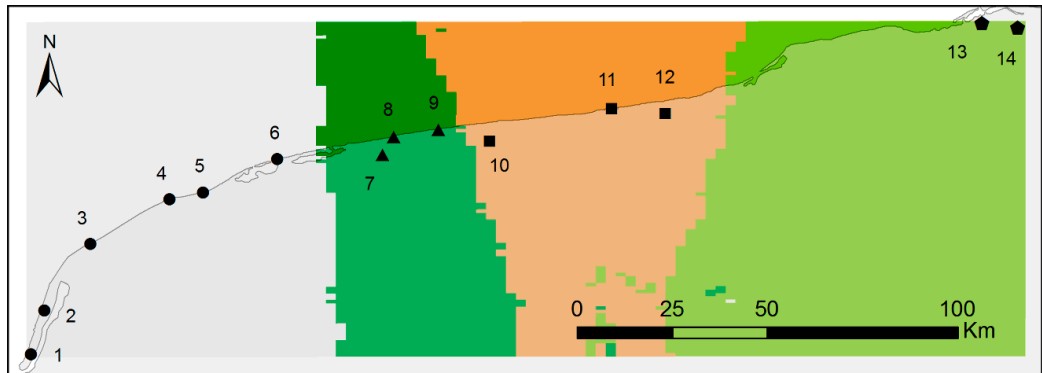

**Figure 4.** Genetic populations identified through Voronoi diagrams obtained in Geneland using the Bayesian grouping method. Four colors and four distinct symbols represent a different genetic population of *Campylorhynchus yucatanicus* on the northern coast of the Yucatan Peninsula, Mexico. Sampling sites are (1) southwest Celestún, (2) northeast Celestún, (3) El Palmar, (4) west Sisal, (5) east Sisal, (6) Chuburná, (7) Capilla, (8) west Chixchulub, (9) San Benito, (10) Xcambó, (11) Santa Clara, (12) Dzilam, (13) west Ría Lagartos, and (14) east Ría Lagartos.

The AMOVA analysis indicated that 12% of total genetic variation was associated with differences between populations defined a priori, while only 3% was attributed to variation between sites within populations. The difference between individuals gave the remainder of the variation (Table A4).

In general, the genetic population of *C. yucatanicus* determined by Geneland analysis with less diversity comprised the Ría Lagartos sites and the westernmost population included sites from Celestún to Chuburná, located west of Progreso city (Table 2). This pattern was also shown with Na, He, and the Shannon Diversity Index (I). Regarding the inbreeding index (F), the Ría Lagartos group had more inbreeding than the western Chuburná sites.

**Table 2.** Genetic diversity $\pm$ standard error in four genetic populations of *C. yucatanicus* identified in the northern Yucatan Peninsula, Mexico. Sample size (N), allele richness (Na), Inbreeding Index (F), Shannon diversity index (I), expected heterozygosity (He), and standard error (SE).

| Genetic Population (Sites) | N | Na $\pm$ SE | F $\pm$ SE | I $\pm$ SE | He $\pm$ SE |
|---|---|---|---|---|---|
| Celestún-Chuburná group (1–6) | 65 | 4.17 $\pm$ 0.60 | 0.05 $\pm$ 0.03 | 0.81 $\pm$ 0.13 | 0.43 $\pm$ 0.07 |
| Chixchulub group (7–9) | 24 | 3.83 $\pm$ 0.79 | 0.10 $\pm$ 0.08 | 0.89 $\pm$ 0.19 | 0.48 $\pm$ 0.08 |
| Xcambó-Dzilam group (10–12) | 28 | 3.33 $\pm$ 0.49 | 0.11 $\pm$ 0.05 | 0.93 $\pm$ 0.16 | 0.52 $\pm$ 0.08 |
| Ría Lagartos group (13–14) | 13 | 2.50 $\pm$ 0.43 | 0.07 $\pm$ 0.01 | 0.56 $\pm$ 0.19 | 0.31 $\pm$ 0.11 |

According to the results from DIYABC RF, the scenario with the highest support (number of RF votes = 41%, posterior probability = 0.496) was scenario 3 (recent bottleneck). According to the posterior distribution of the parameters (Table A5), an ancestral population with an effective size of 44,397.8 individuals (95% CI = 15,075.1–94,751.9) contracted to 1293.24 (95% CI = 363.5–3053.45) individuals approximately 7829 years ago (95% CI = 2218.14–18,108) (assuming a two-year generation time).

### 3.3. Landscape Composition and Genetic Diversity: Node Level

We used 12 models to analyze the influence of landscape structure and habitat fragmentation on genetic diversity (Table 3). The models, including those for the effect of disturbed habitat proportion (CA2) and fragment type diversity (SDI), had a certain probability of being plausible ($2 < \Delta AIC < 4$). However, these models had a low probability of explaining the data ($\Sigma wiCA2$, SDI = 0.14), and regressions were not significant. The best and equally competitive models ($\Delta AIC < 2$) were those that included proportion of suitable habitat (CA1) and index of equitability or fragment uniformity (SEI). These models had a probability of explaining the genetic diversity in the sampling sites, measured by He, of 54% ($\Sigma wiCA1$, SEI = 0.54), although each model by itself weighed less than 30%. The rest of the models did not perform well since their regression coefficients were low and *p* values were greater than 0.05. Proportion of suitable habitat (CA1) had a positive effect on He ($\beta = 0.1$, He = 0.1CA1–0.5). Likewise, the uniformity of fragment types (SEI) in the landscape was positively related to genetic diversity in sampling sites ($\beta = 1.5$, He = 1.5CA1–0.2) (Table 3).

**Table 3.** Models proposed to examine the relationship between the expected heterozygosity (He) of *C. yucatanicus* in the northern Yucatan Peninsula Mexico and landscape structure and composition. The table shows the Shannon Equity Index (SEI), suitable habitat proportion (CA1), fragments diversity index (SDI), disturbed habitat proportion (CA2), risk index for the proximity of human settlements (PA), fragments edge density of suitable habitat (ED1), fragments average shape index of suitable habitat (MSI14), distance to human settlements (SA), distance to road (SC), fragments number of suitable habitat (NumP14), and average size of fragments of suitable habitat (MedPS14). Plausible models (*) are ordered according to ΔAICc values: N: observations number, k: parameters number, logLik: logarithm of likelihood; AICc: Akaike Information Criterion, ΔAICc: the difference between the AIC value of the best model and the AIC of the remaining models, LIKAIC: Likelihood Akaike Information Criterion, ωi: Akaike's weight.

| Model | N | K | logLik | AIC | AICc | ΔAICc | LIKAIC | ωi | R | $R^2$ | F | p |
|---|---|---|---|---|---|---|---|---|---|---|---|---|
| SEI | 14 | 3 | 14.79 | −23.59 | −21.19 | 0.00 | 1.00 | 0.35 | 0.30 | 0.24 | 5.19 | 0.04 * |
| CA1 | 14 | 3 | 14.19 | −22.37 | −19.97 | 1.22 | 0.54 | 0.19 | 0.24 | 0.18 | 3.76 | 0.04 * |
| SDI | 14 | 3 | 13.48 | −20.97 | −18.57 | 2.62 | 0.27 | 0.09 | 0.16 | 0.09 | 2.25 | 0.16 |
| CA2 | 14 | 3 | 12.82 | −19.64 | −17.24 | 3.95 | 0.14 | 0.05 | 0.07 | 0.00 | 0.96 | 0.35 |
| PA | 14 | 3 | 12.44 | −18.87 | −16.47 | 4.72 | 0.09 | 0.03 | 0.02 | −0.06 | 0.28 | 0.61 |
| ED1 | 14 | 3 | 12.39 | −18.79 | −16.39 | 4.80 | 0.09 | 0.03 | 0.02 | −0.07 | 0.20 | 0.66 |
| MSI14 | 14 | 3 | 12.33 | −18.67 | −16.27 | 4.92 | 0.09 | 0.03 | 0.01 | −0.07 | 0.10 | 0.76 |
| log10(SA) | 14 | 3 | 12.33 | −18.66 | −16.26 | 4.93 | 0.08 | 0.03 | 0.01 | −0.08 | 0.09 | 0.77 |
| SC | 14 | 3 | 12.30 | −18.60 | −16.20 | 4.99 | 0.08 | 0.03 | 0.00 | −0.08 | 0.03 | 0.86 |
| CA1 + PA | 14 | 4 | 12.83 | −17.66 | −13.21 | 7.98 | 0.02 | 0.01 | 0.08 | 0.00 | 0.98 | 0.34 |
| CA1 + MSI14 | 14 | 4 | 12.75 | −17.50 | −13.05 | 8.14 | 0.02 | 0.01 | 0.06 | −0.01 | 0.83 | 0.38 |
| NumP14 + MedPS14 | 14 | 4 | 12.33 | −16.66 | −12.21 | 8.98 | 0.01 | 0.00 | 0.01 | −0.08 | 0.09 | 0.77 |

### 3.4. Genetic Distances and Landscape Resistance

Genetic differentiation, measured by $F_{ST}$ mod, was strongly related to the Euclidean distance between sampling sites ($R^2 = 0.45$, $p = 0.001$, Figure 5A). We also found a significant relationship between $F_{ST}$ mod and estimated landscape resistance ($p = 0.02$, Figure 5B). Both resistance and the Euclidean distance between sites had a positive effect on the $F_{ST}$ mod, although the first variable showed a lower regression coefficient ($R^2 = 0.1$). It is interesting to note that in the first range of landscape resistance values (3 < logResistance > 5), there seemed to be a stronger relationship between resistance and the $F_{ST}$ mod.

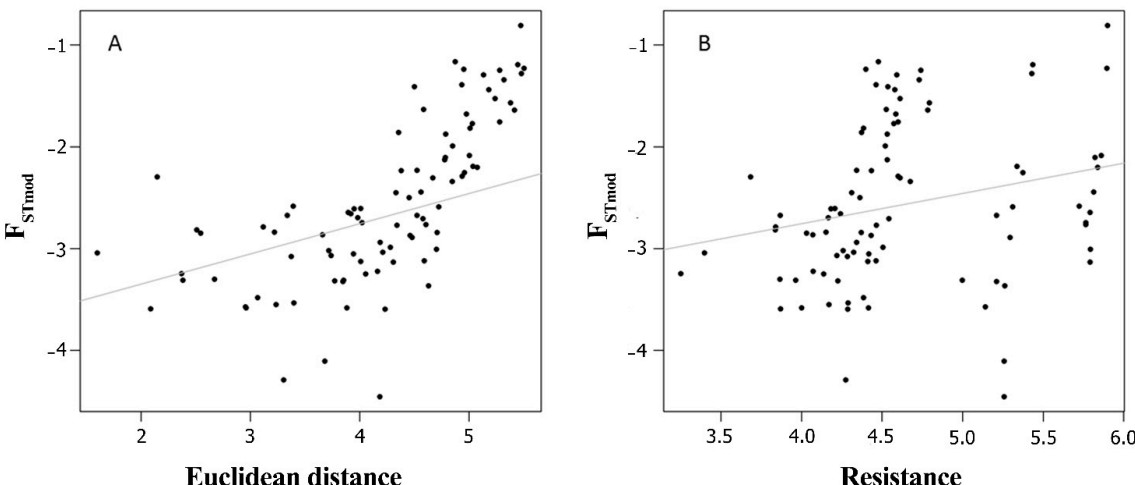

**Figure 5.** Estimated relationships between genetic differentiation ($F_{ST}$ mod) and geographic distance ((**A**), $R^2 = 0.45$, $p = 0.001$) and $F_{ST}$ mod and landscape resistance ((**B**), $R^2 = 0.1$, $p = 0.02$) for *C. yucatanicus* in the northern coast of the Yucatan Peninsula, Mexico in the field during surveys from 2015 to 2016. Values correspond to a logarithmic modification of data.

## 4. Discussion

### 4.1. Genetic Diversity

The genetic diversity, measured as He, of *C. yucatanicus* ranged from low to moderate values along the sites sampled. We identified four genetic groups, which can be assumed to be populations as they were genetically diverse and differentiated by isolation. This was probably promoted by habitat fragmentation mainly due to human activities. The gene flow among populations seemed restricted by habitat loss, promoting resistance to the individuals' movement along the coast.

In general, the low genetic diversity of *C. yucatanicus* (He < 0.50) followed the expected values for threatened bird species, with restricted distribution and relatively small populations [62–64]. However, our results contrast with those found in *C. brunneicapillus* populations with a higher level of genetic diversity (He > 0.50), which is a least-concern species [36]. If the genetic diversity of the identified populations of *C. yucatanicus* decreases and the isolation increases in the next few generations, the species will soon be in rapid decline and on the verge of potential extinction.

### 4.2. Genetic Structure

Through this research project, we found that *C. yucatanicus* currently has even more restricted distribution to the northern coastline of the Yucatan Peninsula, where populations are not homogeneous; instead, there are discrete groups of individuals (probably families), with the presence of four genetic groups which can be considered as populations differentiated probably by habitat loss and geographic distance. The two geographically extreme sites of the *C. yucatanicus* current distribution were the least diverse in terms of heterozygosity and with low genetic interchange, meaning isolated populations and smaller population sizes, with both populations likely being driven to an endogamy process [65] which, in turn, represents increased homozygosity and the expression of deleterious alleles.

The individuals of southwest Celestún are probably confined to a small peninsula and separated to the east by the town of Celestún and to the south by the water of Ría Celestún. The Celestún site has also been affected by occasional fires [66], which can cause habitat loss and decreases in population size, promoting low genetic diversity by a bottleneck, besides the limited influx of new alleles as a result of the presence of Celestún. On the other hand, the individuals of Ría Lagartos inhabit sites of secondary vegetation in intensive and constant use for livestock, probably reducing their survival and reproduction rates. Our DIYABC RF results also indicate that *C. yucatanicus* has experienced a recent population decline, which is probably associated with the Holocene climate and vegetation changes.

The genetic structure of *C. yucatanicus* along the coastline seems to be determined by geographical distance and physical barriers, such as human settlements or unsuitable habitats separating the four genetic populations (Figure 4). The current habitat of *C. yucatanicus* is discontinuous within a complex and structured landscape matrix [61]. This scenario limits connectivity among populations [60,67,68]; therefore, the *C. yucatanicus* populations are genetically structured. Maintaining connectivity among populations will be essential to allow the natural movement of individual birds between sites to preserve the species and its evolution. Some measures of genetic conservation should be addressed in the short term to prevent possible adverse effects of inbreeding depression and, in the long term, to allow the species to adapt and evolve in response to changing environmental conditions [69].

We found that highly populated human settlements on the coast of the Yucatan Peninsula, such as Puerto Progreso, constitute significant induced barriers for *C. yucatanicus* populations and consequently limit the gene flow, which could have implications for the viability of its populations. This anthropic new landscape element seems to be a factor that defines the genetic differentiation of the Celestún-Chuburná group from the rest of the populations. The coastal villages of San Bruno and San Benito are small, with total populations of 45 and 61 people, respectively [70], and could offer resistance to the flow of individuals between the populations of the Xcambó-Dzilam and Celestún-Chuburná

groups. This may be due to the location of Laguna Rosada, to the south of these towns, which could also present limited connectivity between these sites by adding both effects. It has been shown that urbanization is a crucial factor in isolation between patches of habitat in other species of birds and mammals, which causes a strong genetic structure [37,71,72]. Finally, the population of Ría Lagartos is separated from the rest of the sampled sites by an area of unsuitable habitat, limiting free gene flow between the Ría Lagartos group and the nearest Xcambó-Dzilam group.

Occasionally, individuals of *C. yucatanicus* can be found foraging near houses and coastal villages [73], indicating a certain tolerance of human presence. However, anthropic areas used by local people as gardens, backyards, or plots that conserve elements of suitable vegetation and have plant structures can help the species to utilize these resources, perhaps using them as a corridor between populations. On the other hand, sites that are disturbed or that are close to human settlements are more exposed to wildlife diseases (e.g., avian influenza, West Nile Virus, and malaria from poultry) and a high rate of predation by introduced animals (e.g., cats, goats) [74], and they are not suitable as reproduction sites as this species only nests in thorn scrub forests and mangroves [31]. It is imperative to conduct studies on population recruiting, survival, and dispersion within different populations.

### 4.3. Landscape Composition and Genetic Diversity

The relationship between the genetic diversity *C. yucatanicus* and the proportion of suitable habitat and current landscape configuration was significant. Coastal urban development on the Yucatan coast has grown considerably [75], and the habitat of this bird species has been highly affected [33]. These processes are relatively recent. Analyses of these relationships at two different levels, nodes and matrices, allowed us to demarcate habitat loss and fragmentation effects. We recommend that future research incorporate variables that describe changes in landscape configuration over time and add confusion matrices as an analytical method to differentiate both effects [76].

### 5. Implications for Conservation

We detected local extinctions in, for example, Isla Arena and Costa Sur. There are historical reports of presence from GBIF, e-Bird, iNaturalist, and Vargas-Soriano et al. [31]; however, during the fieldwork and a year later, it was ratified that individuals of *C. yucatanicus* were no longer found there. Thus, it is necessary to implement actions to prevent local extinctions.

The results obtained in this study will allow for the identification of focal populations of *C. yucatanicus*, which constitute priorities for decision-makers in the future management plans of Natural Protected Areas (ANPs in Spanish) in the Yucatan Peninsula, Mexico. For example, populations of the Celestún-Chuburná and Ría Lagartos groups with less genetic diversity require more attention, which is worrisome because they are closest to the Biosphere Reserves. This study finds that current conservation strategies within the Mexican ANPs are insufficient and that conservation of *C. yucatanicus* is not conceivable only within the ANPs. Establishing biological corridors with private and state lands is necessary to preserve the *C. yucatanicus* populations. It is crucial and urgent to design management plans focused on improving the connectivity of landscapes inside and outside ANPs [77,78] on the northern coast of the Yucatan Peninsula to maintain the connectivity and diversity of populations and ecosystems.

The Xcambó-Dzilam population has greater genetic diversity and, therefore, constitutes the largest genetic reservoir of the species and a key element in the conservation of *C. yucatanicus* in its natural habitat. The protection of suitable habitats in these sites is fundamental to maintaining species genetic variability and can be a starting point for implementing strategies of functional landscape connectivity in the Yucatan Peninsula. In addition, individuals from this population could be the best option if future translocation programs are needed to increase genetic variability in populations and sites that may require it [79]. In the near future, translocation could be a solution to reduce the risk of

inbreeding depression and reestablish the genetic variability among different populations, favoring species persistence in the long term [80,81], together with land-use programs that incorporate biological corridors and protect additional suitable habitats.

Planning landscape management for species requires detailed knowledge of species habitat use, adequate habitat availability in each site, and matrix permeability between sites [2,82]. Identifying and locating elements in the landscape acting as barriers and corridors is fundamental for such planning [83]. This information should be complemented by patterns of individual movement to document habitat use and occupied areas and estimate gene flow among populations of the focal species to design strategies to maintain the original connectivity between them [84]. In this study, we identified elements that can constitute barriers for *C. yucatanicus*, such as the city of Progreso. Urban planning must consider these elements [85]. We suggest conserving strips of original coastal shrubland between the coast and internal or salt lagoons. A praiseworthy alternative is to encourage gardens and backyards that preserve elements of the original habitat that can function as corridors in place of bare sandy soils. Livestock areas can support some small populations of *C. yucatanicus* while maintaining a certain proportion of elements of the original habitat. It is necessary to promote friendly livestock management with the environment [86] as an alternative for the most disturbing ranches in the Yucatan Peninsula. In addition, it is helpful to promote other economic alternatives to local communities could be explore, among them nature tourism-oriented activities, bird watching, and hiking [87].

We suggest that based on the results of this study and that by Serrano-Rodríguez et al. [61], *C. yucatanicus* be changed from the "near threatened" to the "endangered" category of IUCN-BirdLife International. As suggested by SEMARNAT [34], we considered that criterion B considers an extension less than 5000 km$^2$ and that we estimated a potential distribution of this species to be 2711 km$^2$ [62], including a severely fragmented distribution with only four genetic populations (Criterion B-1-a). Therfore, we propose that the Mexican government, through the CONANP (Comisión Nacional de Áreas Naturales Protegidas), should include *C. yucatanicus* as soon as possible in the recovery of species at risk program (known in Spanish as "PROREST: Programas de Protección y Restauración de Ecosistemas y Especies Prioritarias") in Mexico.

**Author Contributions:** Conceptualization, A.S.-R., G.E.-S., S.M.-M., E.E.I.E. and L.R.-M.; Data curation, A.S.-R.; Formal analysis, A.S.-R., G.E.-S., A.G.R., S.M.-M., E.E.I.E., L.R.-M. and A.H.P.-V., Funding acquisition A.S.-R., G.E.-S and E.E.I.E., Investigation A.S.-R. and G.E.-S., Methodology, A.S.-R., G.E.-S., A.G.R., S.M.-M., E.E.I.E. and L.R.-M., Project administration, A.S.-R. and G.E.-S., Resources A.S.-R., G.E.-S and L.R.-M., software, A.S.-R., Supervision A.S.-R., G.E.-S., A.G.R., S.M.-M., E.E.I.E. and L.R.-M., Visualization, A.S.-R., Writing—original draft, A.S.-R. and G.E.-S., Writing— review & editing, A.S.-R., G.E.-S., A.G.R., S.M.-M., E.E.I.E., L.R.-M. and A.H.P.-V. All authors have read and agreed to the published version of the manuscript.

**Funding:** El Colegio de la Frontera Sur supported this work. CONACYT provided financial support in the form of a scholarship for Anay Serrano-Rodríguez (#308491) and fellowships for G.E.-S. (#21467) and S.M.-M (#217950). This work was also supported by the Small Grants of The Rufford Foundation (A.S.-R. 17373-1 and ASR 23522-2), Cornell Lab of Ornithology seed grants, IIES (Instituto de Investigaciones en Ecosistemas y Sustentabilidad), Idea Wild, and Birder Exchange.

**Institutional Review Board Statement:** It does not apply.

**Data Availability Statement:** The data generated for this manuscript are in the tables and the appendix. Additional information to that cited here may be requested from the authors.

**Acknowledgments:** Thanks to SEMARNAT, SEDUMA, and CONANP for permitting access to protected areas and sample collection (permits SGPA/DGVS/007765/15, SGPA/DGVS/11088/16, and SGPA/DGVS/002465/18). Thanks to Guillermo Castillo, José España, Barbara MacKinnon H., Alexander Dzib, and ANP's workers for field assistance and information sharing. We are grateful to Maricela García Bautista from the Genetic Laboratory at Ecosur-San Cristóbal de las Casas for her orientation on the first steps of the genetic analyses. Thanks to Ricardo Gaytán Legaria, Goretty Mendoza, Libny Ingrid Lara-De La Cruz, and other IIES (Instituto de Investigaciones en Ecosistemas y

Sustentabilidad) students for their help. Thanks to Paul Sosa Moya for the graphic abstract ilustrations. Anonymous reviewers improved an early draft of this manuscript.

**Conflicts of Interest:** The authors declare no conflict of interest.

## Appendix A

There are six tables on (A) microsatellite loci information, (B) prior distribution of the models used in the ABC-RF analyses, (C) assignment probability (%) of sampling sites to four genetic groups identified in Structure Harvester (K = 4) of *C. yucatanicus* in the northern Yucatan Peninsula, Mexico, (D) a summary analysis of molecular variance (AMOVA) with the percentage of variation among genetic groups identified a priori with Geneland between sites in the same group and between individuals of the *C. yucatanicus* in the northern Yucatan Peninsula, Mexico, (E) posterior distribution of the best-fitted model of *C. yucatanicus'* historical demographic changes under scenario 3 (recent bottleneck), and (F) models proposed to examine the relationships between expected heterozygosity y and landscape structure and composition.

This section contains supplemental Information for the effects of anthropogenic habitat fragmentation on the genetic connectivity of the threatened endemic *Campylorhynchus yucatanicus* (Aves, Trogloditydae) in the Yucatan Peninsula, Mexico.

**Table A1.** Microsatellite loci information used to analyze the genetic diversity of *Campylorhynchus yucatanicus* in the north tip of the Yucatan Peninsula, Mexico. Total number of alleles (AT) and alignment temperature (TA). 1 Barr et al. (2015).

| ID | Locus1 | Repetition Units | Primer Sequence (5′-3′) | AT | Length (pb) | TA (°C) |
|---|---|---|---|---|---|---|
| Locus1 | CACW3-01 | (ATT)5G(TTA)4(TTG)6TTATTG(TTGTTA)3(TCA)9 | F: ACTGTTCACCCTTGGACCTG<br>R: TGTCTGGAAACCACTGAAGAAC | 6 | 168–188 | |
| Locus2 | CACW3-03 | (CTA)5CTG(CTA)8(ATA)10 | F: TCCTGAAATGTAATTCAGACACC<br>R: CAGAGTGCTACTTAAATTGATTCTTTC | 5 | 259–279 | 57.6 |
| Locus3 | CACW3-05 | (TGT)5 | F: GATGCATATTGTCAGAGTTCCAC<br>R: CTGGACTGAGCTAACAAATGATG | 5 | 131–149 | 57.6 |
| Locus4 | CACW3-11 | (ATA)5(AAC)6AAT(AAC)4(AAT)3AG(TAA)4 | F: TTCTCCTCCCTCTACCTCCTTT<br>R: GTGACAACAGAAAATTCCCTTTA | 8 | 180–204 | 54 |
| Locus5 | CACW4-01 | (GTAT)6GAATCTG(TCTA)11 | F: TTTTGCCTAATAAACTGGCTGAC<br>R: CACAGAACCACAACCTACATGG | 3 | 122–133 | 54 |
| Locus7 | CACW4-04 | (TCTA)14 | F: TCTCACGTCTTACCATCCTGTG<br>R: TTGATACTTGAAACTCTCCTTCTGTC | 5 | 241–257 | 57.6 |
| Locus9 | CACW4-09 | (GATG)22 | F: GCTAACTGAAAGGGATTGTTGG<br>R: TTTCTGGCATGTTTCCTGTC | 5 | 92–116 | 59 |

**Table A2.** Prior distribution for each parameter of the models used in the ABC-RF analyses. N = effective population size, T = time (in years).

| Parameters | Type | Prior |
|---|---|---|
| N1 | N | UN~[10-100,000–50,000-0] |
| N2 | N | UN~[10-100,000–50,000-0] |
| N3 | N | UN~[10-100,000–50,000-0] |
| N4 | N | UN~[10-100,000–50,000-0] |
| N5 | N | UN~[10-100,000–50,000-0] |
| t1 | T | NO~[500–13,000–2500-1] |
| t2 | T | NO~[13,000–65,000–7,500-1] |

**Table A3.** Assignment probability (%) of sampling sites to four genetic groups identified in Structure (K = 4) of *Campylorhynchus yucatanicus* in the northern Yucatan Peninsula, Mexico.

| ID | Sites | N | K1 | K2 | K3 | K4 |
|----|-------|---|-----|-----|-----|-----|
| 1 | Southwest Celestún | 7 | 13.21 | 30.64 | 24.31 | 31.86 |
| 2 | Northeast Celestún | 15 | 10.87 | 30.36 | 27.31 | 31.44 |
| 3 | El Palmar | 10 | 19.81 | 27.72 | 24.89 | 27.62 |
| 4 | West Sisal | 10 | 9.21 | 30.36 | 29.61 | 30.85 |
| 5 | East Sisal | 13 | 15.69 | 28.04 | 28.95 | 27.34 |
| 6 | Chuburná | 15 | 11.36 | 31.49 | 24.71 | 32.44 |
| 7 | Capilla | 8 | 16.9 | 25.53 | 32.91 | 24.68 |
| 8 | Chixchulub | 4 | 26.12 | 24.16 | 26 | 23.7 |
| 9 | San Benito | 12 | 16.28 | 27.46 | 28.02 | 28.25 |
| 10 | Xcambó | 10 | 26.54 | 23.39 | 28.07 | 21.98 |
| 11 | Santa Clara | 10 | 18.62 | 25.37 | 31.42 | 24.61 |
| 12 | Dzilam | 12 | 46.11 | 17.8 | 18.71 | 17.37 |
| 13 | West Ría Lagartos | 4 | 68.63 | 10.57 | 11.27 | 9.53 |
| 14 | East Ría Lagartos | 10 | 72.37 | 9.33 | 9.4 | 8.91 |

**Table A4.** Summary Analysis of molecular variance (AMOVA), percentage of variation among genetic groups identified *a priori* with Geneland, between sites in the same group, and between individuals of the *Campylorhynchus yucatanicus* in the northern Yucatan Peninsula, Mexico.

| Source | Df | SS | MS | Est. Var. | % |
|--------|-----|---------|--------|-----------|-----|
| Among Populations *a priori* | 3 | 38.837 | 12.946 | 0.190 | 12 |
| Among Sites | 10 | 20.629 | 2.063 | 0.043 | 3 |
| Among Indiv | 116 | 147.649 | 1.273 | 0.000 | 0 |
| Within Indiv | 130 | 179.000 | 1.377 | 1.377 | 86 |
| Total | 259 | 386.115 | | 1.610 | 100 |

**Table A5.** Posterior distribution for each parameter of the best-fitted model of *Campylorhynchus yucatanicus'* historical demographic changes under scenario 3 (recent bottleneck) based on a training reference table of 100K simulations and 500 RF decision trees.

| Parameter | Mean | Median | Quantiles 0.05 | Quantiles 0.95 |
|-----------|---------|---------|----------------|----------------|
| N1 | 1293.24 | 1032.23 | 363.5 | 3056.45 |
| N2 | 44397.8 | 37826.6 | 15075.1 | 94751.9 |
| t1 | 7829.86 | 5750.76 | 2218.14 | 18108 |

**Table A6.** Models proposed to examine relationship between expected heterozygosity (He) of *Campylorhynchus yucatanicus* in the northern Yucatan Peninsula Mexico and landscape structure and composition.

| Model | |
|-------|--|
| SEI | Shannon Equity Index |
| CA1 | Suitable habitat proportion |
| SDI | Fragments diversity index |
| CA2 | Disturbed habitat proportion |
| PA | Risk index for proximity of human settlements |
| ED | Fragments edge density of suitable habitat |
| MSI14 | Fragments average shape index of suitable habitat |
| log10(SA) | Log10 distance to human settlements |
| SC | Distance to road |
| CA1 + PA | Combined effect of Suitable habitat proportion and Risk index for proximity of human settlements |
| CA1 + MSI14 | Combined effect of Suitable habitat proportion and Fragments average shape index of suitable habitat |
| NumP14 + MedPS14 | Combined effect of fragments number of suitable habitat and average size of fragments of suitable habitat |

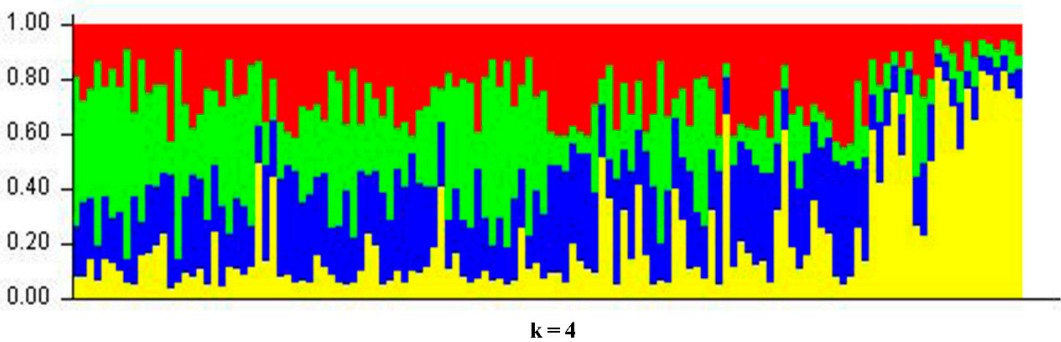

**Figure A1.** Results of structure cluster analysis.

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
