# Peer review of "Effects of Anthropogenic Habitat Fragmentation on the Genetic Connectivity of the Threatened and Endemic Campylorhynchus yucatanicus (Aves, Trogloditydae) in the Yucatan Peninsula, Mexico"

_diversity, doi:10.3390/d14121108_

Round 1

Reviewer 1 Report

Serrano-Rodríguez et al., analyzed the genetic diversity and structure of Campylorhynchus yucatanicus based on seven nuclear microsatellite loci. Results did not obtain significant divergence among populations (Four genetic clades were shown in each of sampling populations). This could caused by narrow sampling region or rough molecular data. I suggest authors employ more loci (at lest more than 15 SSR loci) or genomic approaches (i.e. RADseq, GBSseq) to investigate the genetic structure. Also, authors would be better to collect more populations across the species distribution region.

In the part of Genetic structure analysis, it is better to analyze the demographical history (i.e. ABC-RF software) to well understand the evolutionary mechanism of this endangered species.

Minor concerns:

Figure 1: Campylorhynchus. Yucatanicus ---> Campylorhynchus yucatanicus

Table 3: Xcambó-Dzilam group (1-9) ---> Xcambó-Dzilam group (7-9)

Figure 4: Please add the values of r and p in the picture.

Author Response

MDPI _ Reply review report Reviewer 1

Reviewer 1 comments in bold, answers in green.

1) Serrano-Rodríguez et al., analyzed the genetic diversity and structure of Campylorhynchus yucatanicus based on seven nuclear microsatellite loci. Results did not obtain significant divergence among populations (Four genetic clades were shown in each of sampling populations). This could caused by narrow sampling region or rough molecular data. I suggest authors employ more loci (at lest more than 15 SSR loci) or genomic approaches (i.e. RADseq, GBSseq) to investigate the genetic structure. Also, authors would be better to collect more populations across the species distribution region.

     The genetic variability of Campylorhynchus yucatanicus, an endemic bird to the Yucatan Peninsula, has been addressed in this study for the first time, so there is no history of described polymorphic genetic markers specific to the species. We decided to try ten markers used in C. brunneicapillus (Barr et al. 2015) to optimize resources and time. We were unsuccessful in amplifying three of them, and finally, seven effective markers were the ones we used in this study. Species with high commercial value or conspicuous species have libraries of genetic markers described; for example, at least 428 microsatellite markers have been defined for Oncorhynchus mykiss (Rexroad et al. 2005). However, other non-commercial species, such as C. yucatanicus, small in size, endemic, and threatened, are little studied, and we have less information about them. In any case, in the literature, we find valuable cases that characterize genetic diversity in various species with less than 15 genetic markers (Garcia-Moreno et al. 1996, Castella and Ruedi 2000, Thulin et al. 2002, Castro et al. 2006). A similar number of microsatellite markers have been sufficient to describe the population genetic structure of ecologically complex species and to suggest concrete management actions (Florin & Höglund 2008) and even fewer (Ball et al. 2000). Of course, having more markers is always better, but developing new markers by SSR would be costly in time and money, which we did not have for the study. We do not have the option to restart the study from the sampling again.

     The analyses mentioned by the reviewer, such as RADseq and GBS, are now very popular since they are very informative. Still, they are relatively expensive, and in that case, it would involve redoing the work to obtain samples, which is impossible right now.

     Regarding the sample size and geographic location of the individuals collected, we ensure that our sample is highly representative of the historical and potential distribution of the species (Paynter 1955, Howell & Webb 1995, Serrano-Rodríguez et al. 2017, Serrano-Rodríguez et al. 2018). Campylorhynchus yucatanicus is a species with distribution restricted to a narrow strip of vegetation associated with the north coast of the Peninsula. We sampled from the westernmost to the easternmost end of the distribution and covered its entire distribution.

2) In the part of Genetic structure analysis, it is better to analyze the demographical history (i.e. ABC-RF software) to well understand the evolutionary mechanism of this endangered species.

     We found this suggestion interesting; however, this is the first study on the diversity and genetic structure of the Yucatán wren (C. yucatanicus), and the possible causes of diversity and structure are just beginning to be identified in this work. Therefore, for now, it is appropriate to identify the structure with Bayesian statistics based on a model implemented in Structure and Geneland. For another study, the use of microsatellite or NGS markers could be increased; more structure models can be proposed and weighted with ABC-RF software, which will give us information about demographic history that is not the objective of the present study.

Minor changes:

Figure 1: Campylorhynchus. Yucatanicus ---> Campylorhynchus yucatanicus

Campylorhynchus yucatanicus was corrected.

Table 3: Xcambó-Dzilam group (1-9) ---> Xcambó-Dzilam group (7-

9).

The new version is table 4, and Xcambó-Dzilam group (7-9) was corrected.

Figure 4: Please add the values of r and p in the picture.

Values of r and P were added in the figure caption.

Literature

Castella, V., & Ruedi, M. 2000. Characterization of highly variable microsatellite loci in the bat Myotis myotis (Chiroptera: Vespertilionidae). Molecular Ecology, 9(7): 1000–1002. doi:10.1046/j.1365-294x.2000.00939-6.x

Garcia-Moreno, J.; Roy, M.S.; Geffen, E.; Wayne, R. K. 1996. Relationships and genetic purity of the endangered Mexican wolf based on analysis of microsatellite loci. Conservation Biology, 10:396-405.

Serrano-Rodríguez, A.; Escalona-Segura, G.; Plasencia Vázquez, A.H.; Iñigo Elias, E.E.; Ruiz-Montoya, L. Distribución potencial y conectividad del paisaje: criterios para reevaluar el grado de amenaza de Campylorhynchus yucatanicus (Aves: Troglodytidae). Revista de Biología Tropical (International Journal of Tropical Biology) 2017, 65, 1554-1568. http://dx.doi.org/10.15517/rbt.v65i4.26599.

Rexroad CE 3rd, Rodriguez MF, Coulibaly I, Gharbi K, Danzmann RG, Dekoning J, Phillips R, Palti Y. 2005. Comparative mapping of expressed sequence tags containing microsatellites in rainbow trout (Oncorhynchus mykiss). BMC Genomics. Apr 18;6:54. doi: 10.1186/1471-2164-6-54. PMID: 15836796; PMCID: PMC1090573.

Thulin, C.-G., Gyllenstrand, N., Mccracken, G., & Simberloff, D. 2002. Highly variable microsatellite loci for studies of introduced populations of the small Indian mongoose (Herpestes javanicus). Molecular Ecology Notes, 2(4), 453–455. doi:10.1046/j.1471-8286.2002.00275.x 

Boscherini G., Morgante M., Rossi P., Vendramin G. G., Vicario F. 1994. Detection od DNA polymorphisms in Pinus leucodermis Ant. using random amplification. Forest Genetics 1 (3), 131-137.

Moreno S., Gorgoncena Y., Ortiz J. M. 1997. The use of RAPD markers for identification of cultivated grapevine (Vitis vinifera L.). Scientia Horticulturae (Netherlands) 62 (4), 237-243.

Florin, A.-B., & Höglund, J. 2008. Population structure of flounder (Platichthys flesus) in the Baltic Sea: differences among demersal and pelagic spawners. Heredity, 101(1), 27–38. doi:10.1038/hdy.2008.22

Ball A. O., Sedberry G. R., Zatco M. S., R. W. Chapman, y J. L. Carlin. 2000. Population structure of the wreckfish Polyprion americanus determined with microsatellite genetic markers. Marine Biology, 137: 1077:1090.

Paynter, R. A. 1955. The Ornithogeography of the Yucatan Peninsula. New Haven, Connecticut: Museum of Comparative Zoology Harvard University.

Howell, S., & Webb, S. 1995. A Field Guide to the Birds of Mexico and Northern Central America.  New York, USA: Oxford University Press.

Serrano-Rodríguez A., Escalona-Segura G., Iñigo Elías E. E., Serrano Rodríguez A. , Uriostegui J. M., and Montes De Oca Aguilar A. C. 2018. Potential distribution and climatic niche of seven species of Campylorhynchus (Aves, Troglodytidae): conservation implications for C. yucatanicus. The Wilson Journal of Ornithology, 130(1): 13-22. https://doi.org/10.1676/16-101.1

Reviewer 2 Report

In order to detect how land use change can affect the genetic connectivity of a threatened endemic bird Campylorhynchus yucatanicus in the Gulf of Mexico, the authors collected 140 samples from 14 localities and used seven nuclear microsatellite loci to describe the current structure and genetic diversity of the populations. They found human settlements might limit the connectivity between different sites due to the ongoing land use changes. Therefore, future coastal development should take care of the bird conservation. The topic falls within the scope of Diversity. Some concerns need to be taken care below:

1)       The content of findings in abstract is much limited, especially compared with that of background description.

2)       The part of discussion can be more focused. Listing subtitles would work better than current form.

3)       According to Figure 3, significant difference existed between genetic population of Campylorhynchus yucatanicus. What are the fundamental reasons? Or say, what the boundary between different colors may present in reality.

4)       Writing can be improved much. The readability needs to be strengthened in my mind, though I am not an English-native speaker. Below, I only listed a very few of the awkward sentences, and I believe there existed more. 

Abstract

“These human-induced processes of land use change can affect the Yucatan Wren (Campylorhynchus yucatanicus), an endemic bird to these vegetation types with a narrow distribution.” The latter half is not clear. 

“We found four genetic populations with Bayesian clustering methods.” Not clear. 

“We suggest some management actions for conservation of this species, and we propose to change the IUCN threat category to "endangered" because of today the species has a more restricted distribution, small population, habitat degradation, loss of connectivity, and loss of genetic variability.” Need to be revised. 

Discussion

They are genetically highly diverse and differentiated by isolation by distance promoted by habitat fragmentation due mainly to human activities. A better presentation is needed. 

In general, the low genetic diversity of C. yucatanicus (He < 0.50) follows the expected for threatened bird species… need to be revised

Author Response

Reviewer 2 comments in bold and answers in blue.

1) The content of findings in abstract is much limited, especially compared with that of background description.

     We made some modifications in the abstract to clarify the doubts of the reviewer without exceeding the limit of words imposed by the journal (200 words).

2) The part of discussion can be more focused. Listing subtitles would work better than current form.

    Subtitles were added to the discussion.

3) According to Figure 3, significant difference existed between genetic population of Campylorhynchus yucatanicus. What are the fundamental reasons? Or say, what the boundary between different colors may present in reality.

     The actual barriers that are likely limiting connectivity reflected in the differentiation between the four groups are explained from line 442. The port and city Progreso are located between the first group (gray area represented by circles in Figure 3) and the second group (dark green zone represented by triangles). Perhaps this anthropic element determines the differentiation between these groups. Between San Benito and Xcambó, there is an area where the favorable vegetation for C. yucatanicus narrows to approximately 50m, limited to the north by the sea and to the south by the coastal road and the floodplains of Laguna Rosada. On that narrow strip are the houses and rest villas of San Bruno and San Benito. The properties found in that area have different management of their outdoor areas; some remove the vegetation entirely, leaving the sand bare or coconut plantation instead of the original vegetation. A combination of natural and anthropic factors could limit connectivity between the second group (orange zone represented by squares), which is explained from line 447 in the manuscript. The fourth group, represented by hexagons in Figure 3, is isolated by an area of unfavorable vegetation; this is explained from line 453.

4)        Writing can be improved much. The readability needs to be strengthened in my mind, though I am not an English-native speaker. Below, I only listed a very few of the awkward sentences, and I believe there existed more.

     We also rephrased some parts of the manuscript to make it more traightforward and asked Greg Budney, a native English speaker and a specialist (Curator of Collections Development, Macaulay Library, Cornell Lab of Ornithology) to check it.

5) Abstract

“These human-induced processes of land use change can affect the Yucatan Wren (Campylorhynchus yucatanicus), an endemic bird to these vegetation types with a narrow distribution.” The latter half is not clear.

     We rewrite it as follow:

     The coastal scrub and dune vegetation complex of the northern Yucatan Peninsula has been affected by coastal human development; this can affect the Yucatan Wren (Campylorhynchus yucatanicus) because it is a species associated with these types of vegetation that have a restricted distribution to the Yucatan Peninsula.

6) “We found four genetic populations with Bayesian clustering methods.” Not clear.

     We rephrased it as:

     Four genetic populations were highlighted by the clustering method implemented in the Genland program.

7) “We suggest some management actions for conservation of this species, and we propose to change the IUCN threat category to "endangered" because of today the species has a more restricted distribution, small population, habitat degradation, loss of connectivity, and loss of genetic variability.” Need to be revised.

     We checked carefully this paragraph and rephrased it as follows:

     We suggest changing the IUCN threat category to endangered as we found loss of genetic variability, in addition to its restricted distribution, small population, habitat degradation, and loss of connectivity.

8) Discussion

       They are genetically highly diverse and differentiated by isolation by distance promoted by habitat fragmentation due mainly to human activities. A better presentation is needed.

     We rephrased it to make it clearer:

     The genetic diversity, measured as He, of C. yucatanicus ranges from low to moderate values along sites sampled. We identify four genetic groups, which can be assumed as subpopulations as they are genetically diverse and differentiated by isolation promoted by habitat fragmentation due mainly to human activities. The gene flow among subpopulations seems restricted by habitat loss, promoting resistance to the individual’s movement along the coast.

Reviewer 3 Report

Review for Diversity: Serrano-Rodríguez et al. “Effects of anthropogenic habitat fragmentation on the genetic connectivity of the threatened endemic Campylorhynchus yucatanicus (Aves, Trogloditydae) in Yucatan Peninsula, Mexico”

The study is on the genetic diversity of an endemic bird species whose coastal habitat has recently suffered from increasing modification and fragmentation through human land use. The authors conclude that the species should be recognized as threatened and highlight differences between subpopulations in genetic diversity. I like how the authors explicitly use their results to formulate management advice for the protection of this species. The structure of the manuscript is fine and the use of the literature appropriate. However, the methods need to be clarified by adding further details and the English needs to be improved throughout. The formatting is messed up in some places (with unnecessary bold font, page wide paragraphs and centre-formatted paragraphs). I append some detailed comments below.

Specific comments

The language could be improved in some places. For example, in the introduction section, rephrase the first sentence “Fragmentation and native habitats loss involve a significant reduction …” as “Fragmentation and loss of native habitats involve a significant reduction …”. Also rephrase the last sentence of the same paragraph.

The first sentence in the methods section is partially in bold font. The formatting of the whole paragraph is wrong.

I do not quite understand the sampling design. In each site there were transects, and vocalizations were played every 100 m along these transects. But how long were these transects (how many stops were there). Also where were the mist nets located? Maybe adding a figure for the experimental design would help.

Why were only 130 out of the 186 individuals captured by mist nets considered in the study? This information is currently lacking. I assume it was either to ensure a balanced sample size among sites or because of a lack of capacity to sequence more.

“We captured individuals marked with a unique combination of colored Darvick leg bands.” This sentence is unclear. Do the authors mean to say that they marked the captured individuals using these bands? At present it reads as if the bands were already present when the birds were captured.

Could you add a table showing the 12 a priori models or hypotheses in the methods section? Define the different factors entered in those. Table 4 goes some way to addr4ess this but is not enough.

Table 2: Add a footnote explaining that the IDs are as in Fig. 1. Also check the formatting. Part of the first row is in bold font (also in Table 3). Maybe list the different measures of genetic diversity in the same order in the table caption as they are presented in the table itself, where F comes before I (the same is true in Table 3). Also add information on what type of variation is given for the different measures (SE? SD? …).

Figure 2: “Campylorhynchus yucatanicus” should be in italics. It would be possible to just state that sampling sites are as in Figure 1.

Table 4: The caption states that “Plausible models are high-lighted in bold …”. However, all models are highlighted in bold in this table, including those with very high P-values.

As the city of Progresso and its role as a barrier is referred to several times it may be useful to show its location in one of the figures. Or to give it at first mention in the text as between sites x and y.

Reference 29: the full stop after the authors and before the title is missing.

The fonts used differ between Appendix A (Arial) and Appendix B (Times New Roman).

Author Response

MDPI _ Reply review report Reviewer 3

Reviewer 3 comments in bold and answer in orange.

1) The language could be improved in some places. For example, in the introduction section, rephrase the first sentence “Fragmentation and native habitats loss involve a significant reduction …” as “Fragmentation and loss of native habitats involve a significant reduction …”. Also rephrase the last sentence of the same paragraph.

     We rewrote the paragraph, and the last sentence was changed this way:

     “A management for growing the size of C. yucatanicus local populations could mitigate these problems [15] but preserving or restoring connectivity between populations could be more beneficial for long-term species persistence.”

2) I do not quite understand the sampling design. In each site there were transects, and vocalizations were played every 100 m along these transects. But how long were these transects (how many stops were there). Also where were the mist nets located? Maybe adding a figure for the experimental design would help.

     To clarify this part of methods we rewrite it as follows:

     Sampling was performed through an intensive survey of individuals throughout the range of the species. To do this, we visit those with historical records of species, those with coastal scrub vegetation, and the coastal dune vegetation complex with mangrove edge. Sites where individuals of the species were no longer found were discarded, so the fieldwork included 14 historical sites.

     Once we were at the sites, we walked variable transects (from 1 to 5km), depending on the extent of vegetation favorable to the species. In total, we walked 1,077.58 kilometers where the species could be found.

     At each site, we first walked the transects looking for C. yucatanicus in the early morning hours, and if we had no sighting records of C. yucatanicus, then we used the recordings every 100 meters for 6 minutes.

     The transects allow us to organize the search and minimize the disturbances in the same place, delimiting distances between the reproductions recorded with claims of a conspecific (playback).

     When we recorded individuals of the species, we set up two adjacent nets of 12m with a mesh of 2.5X 2.5 cm by 3m height. After capturing one or more individuals and taking the necessary data, we removed the nets, released the individual or couple, and continued the search for a new individual along the transect. Keep in mind that this species is territorial, and if we have fixed nets, we would be capturing the same individuals again and again, causing more stress and decreasing the efficiency of the search and capture of individuals.

3) Why were only 130 out of the 186 individuals captured by mist nets considered in the study? This information is currently lacking. I assume it was either to ensure a balanced sample size among sites or because of a lack of capacity to sequence more.

     We excluded individuals with incomplete genetic information during the lab process and juveniles captured with an adult as they move in familiar groups. Thus, we minimized the effect of genetic resemblance through family ties. ThiThis information was added in the manuscript.

4) We captured individuals marked with a unique combination of colored Darvick leg bands.” This sentence is unclear. Do the authors mean to say that they marked the captured individuals using these bands? At present it reads as if the bands were already present when the birds were captured.

     Yes, “We captured individuals and we marked them with a unique combination of colored Darvick leg bands”. They were not individuals previously banded. We adjust the sentence in the manuscript.

5) Could you add a table showing the 12 a priori models or hypotheses in the methods section? Define the different factors entered in those. Table 4 goes some way to addr4ess this but is not enough

     Including the table with the 12 a priori models is redundant for the manuscript, but we included in the supplementary material in the Appendix C.

6) Table 2: Add a footnote explaining that the IDs are as in Fig. 1. Also check the formatting.

     In this case it is not necessary to specify in the table header the locations of the ID (as in Figure 1) because precisely the information is in the second column.

7) Part of the first row is in bold font (also in Table 3). Maybe list the different measures of genetic diversity in the same order in the table caption as they are presented in the table itself, where F comes before I (the same is true in Table 3).

Also add information on what type of variation is given for the different measures (SE? SD? …).

     We corrected the table and give the requested information.

8) Figure 2: “Campylorhynchus yucatanicus” should be in italics. It would be possible to just state that sampling sites are as in Figure 1.

     Campylorhynchus yucatanicus is italics and certainly it can be stated that the sites are the same as in figure 1, but since the figures are several pages apart, we decided better to leave the names. In addition, if they do not occur, for someone who does not know the Yucatán Peninsula can hardly associate the probabilities of formation of the groups with the sites.

7) Table 4: The caption states that “Plausible models are high-lighted in bold …”. However, all models are highlighted in bold in this table, including those with very high P-values.

     It is right, we adjust the table and only the first two lines are in bold now.

8) As the city of Progresso and its role as a barrier is referred to several times it may be useful to show its location in one of the figures. Or to give it at first mention in the text as between sites x and y.

    We cited were Progreso is located in the manuscript and also in Figure 1.

9) Reference 29: the full stop after the authors and before the title is missing.

     We corrected the format.

10) The fonts used differ between Appendix A (Arial) and Appendix B (Times New Roman).

     The format was standardized to Arial font.

Round 2

Reviewer 1 Report

Thanks a lot for your modifications! I really think it is better to employ more SSR loci or RADseq to investigate the genetic structure of the species with such narrow distributions. The present results did not show genetic divergence among these sampling populations using only seven loci. You can develop more SSR loci by EST library, enrichment libraries and high-throughput sequencing. 

Geneland was used to distinguish genetic structure. Demographical history is valuable to assess the evolutionary potential for endangered species. I suggest the authors add these analysis.
